# Exploring Fungicide Sensitivity in Soybean Stem Blight Pathogen *Diaporthe longicolla*, Emphasizing Genetic Variability Impact on Response to SDHI Fungicides Fluopyram and Pydiflumetofen

**DOI:** 10.3390/jof11040292

**Published:** 2025-04-08

**Authors:** Shanshan Chen, Zhanyun Liu, Zhengjie Chang, Yuxin Zheng, Xueyang Wang, Ningwei Li, Zhongqiao Huang, Can Zhang, Xili Liu

**Affiliations:** 1Department of Plant Pathology, China Agricultural University, Beijing 100193, China; 18800172207@163.com (S.C.); liuzhanyun2023@163.com (Z.L.); scauczjczj@163.com (Z.C.); zyx18854807007@163.com (Y.Z.); 17880151175@163.com (X.W.); lingshilnw@163.com (N.L.); huangzhongqiao@cau.edu.cn (Z.H.); seedling@cau.edu.cn (X.L.); 2State Key Laboratory of Crop Stress Biology for Arid Areas, Northwest A&F University, Yangling 712100, China

**Keywords:** soybean stem blight, *Diaporthe longicolla*, fungicide sensitivity, SdhC

## Abstract

*Diaporthe* species are critical plant pathogens that contribute to a disease complex responsible for substantial yield losses in soybean production worldwide. However, reports on the primary *Diaporthe* species causing soybean stem blight and their sensitivity to various fungicides are scarce in China. In this study, a total of 46 *D. longicolla* strains were isolated and identified from diseased soybean stems and rots collected from 14 regions of Heilongjiang province in 2021 and 2022. Among the eight fungicides examined, fludioxonil, mefentrifluconazole, tebuconazole, and azoxystrobin demonstrated effective inhibition for *D. longicolla*, with EC_50_ values < 0.3 µg/mL. Interestingly, the EC_50_ values of *D. longicolla* to two succinate dehydrogenase inhibitors (SDHIs), pydiflumetofen and fluopyram, were 5.47 µg/mL and over 100 µg/mL, respectively. In molecular dynamics simulations, pydiflumetofen exhibited a smaller RMSD, while fluopyram had a higher binding free energy with Sdh proteins compared to pydiflumetofen. This difference may contribute to the higher activity of pydiflumetofen in *D. longicolla*. Further analysis of the electrostatic potential and structural conformations of the binding pocket revealed that pydiflumetofen formed more hydrophobic interactions with SdhC and SdhD and was positioned closer to the SdhD subunit. A mixture of fludioxonil and mefentrifluconazole at a ratio of 1:5, as well as fludioxonil and pydiflumetofen at a ratio of 1:5, exhibited synergistic effects. These findings demonstrated that several fungicides could be utilized to control *Diaporthe* stem blight, and the difference in binding affinity to the Sdh subunit impacts sensitivity to fluopyram and pydiflumetofen.

## 1. Introduction

Soybean [*Glycine max* (L.) Merr.], which originated in China, is a critical oilseed crop and a valuable source of vegetable protein [1,2]. China is the largest soybean consumer and importer globally, and the Chinese government-proposed soybean revitalization plan and subsidized soybean farmers were enacted in 2019 and 2020 [3]. Total area of soybean planting in China have increased recently, and the largest soybean producing province of China is Heilongjiang, accounting for nearly 5 million hectares of soybeans [4,5]. Frequent disease occurrence is an important factor restricting high and stable soybean yields.

*Diaorthe* species can produce cankers, diebacks, root rots, fruit rots, leaf spots, blights, decay, and wilts on an array of plant hosts, encompassing soybean and economically important hosts worldwide [6,7]. In soybeans, *Diaporthe* species are important seed-borne pathogens causing serious seed yield loss [8]. They are responsible for pod and stem blight, attributed to *D. longicolla*, seed decay caused by *D. longicolla*, and stem canker attributed to *D. aspalathi* and *D. caulivora*, respectively [9,10,11]. Recently, stem blight has become more prevalent in soybeans in China, such as in the Huang-Huai-Hai region, with typically 10 to 30% yield loss being associated with stem blight [12,13]. It is unclear whether *Diaporthe* species cause soybean stem blight in Heilongjiang province. Moreover, soybean root rot occurs in conjunction with soybean stem blight in the field, with *Fusarium* species being a major pathogen causing soybean root rot [14,15,16,17].

Currently, chemical fungicides are still the primary measure of controlling soybean disease, dominated by metalaxyl, carbendazim, and fludioxonil. However, based on the Chinese pesticide information network, no chemical fungicides have been registered for controlling soybean stem blight (http://www.chinapesticide.org.cn/) (accessed on 20 May 2024). Due to the frequent occurrence of soybean stem blight, clarification of the sensitivity of different fungicides to *D. longicolla* is required to select suitable seed treatment fungicides for scientifically controlling soybean diseases. Succinate dehydrogenase inhibitors (SDHIs) have been the fastest-growing fungicides in agriculture over the past fifteen years, with important roles in the chemical control of soybean diseases, and are the third largest class of fungicides in the world after quinone outside inhibitors (QoIs) and demethylation inhibitors (DMIs) [18,19].

SDHIs inhibit the reduction of ubiquinone by binding to the active site of complex II (succinate dehydrogenase B, C, or D subunits), inhibiting ATP production [20]. Prior studies have demonstrated that mechanisms of fungicide binding to subunits influence fungicide sensitivity. Variations in the binding sites of pathogens to fungicides can lead to differences in the sensitivity of one pathogen to the same fungicide mode of action for instance, Amiri reported that the anchored SdhC subunit affects the fitness and sensitivity of *Botrytis cinerea* to SDHIs [21]. In Chen’s study, significant differences in sensitivity of *Fusarium asiaticum* causing *Fusarium* head blight to pydiflumetofen were identified between mutants of the anchored SdhB subunit and the anchored SdhC subunit, with EC_50_ values of 0.36 and 60.67 µg/mL, respectively. Mirroring Shao’s study, differences in the sensitivity of *Fusarium graminearum* to pydiflumetofen were also documented [22,23]. The sensitivity and binding modes of *Diaporthe* species to SDHIs like pydiflumetofen remain unknown.

Therefore, this study was performed to (i) clarify *Diaporthe* species causing soybean stem blight in Heilongjiang, (ii) evaluate the sensitivity of *D. longicolla* to seven fungicides, and (iii) examine the mechanism underlying the differences in sensitivity to two SDHI fungicides.

## 2. Materials and Methods

### 2.1. Collection of Diaporthe Species Isolates

In 2021 and 2022, a total of 46 *Diaporthe* isolates were obtained from small pieces of about 0.4 cm^2^ of the dried stems of 14-week-old mature plants across 14 different regions of the Heilongjiang province in China. The isolates were grown in a dark environment on potato dextrose agar (PDA, 200 g/L potato, 20 g/L dextrose, and 15 g/L agar) plates. The culture temperature was maintained at 25 °C. To observe pycnidia, isolates were inoculated onto autoclaved soybean stems and placed on water agar medium (WA, 15 g/L agar and 1 L autoclaved water), and incubated at 25°C with a 12-h light/dark cycle over a period of 28 days. The *Diaporthe* isolates were preliminarily identified according to morphological characteristics like colony appearance and color, conidiophores, and pycnidia (under the microscope) [24]. Following single-spore isolation, the mycelia of each isolate were preserved for long-term storage in 5 mL plastic tubes with PDA medium slants. The tubes were sealed with mineral oil and stored at 14 °C.

### 2.2. DNA Extraction and Identification of Diaporthe Isolates

Based on the culture condition, the colony morphologies were observed on the PDA medium after being cultured for three days. For molecular identification, the DNA of the *Diaporthe* isolates was extracted using the cetyltrimenthyl ammonium bromide (CTAB) extraction method [25]. The isolated DNA samples were validated through PCR analysis using primers targeting the internal transcribed spacer (ITS), elongation factor (*EF1-α*) genes, partial beta-tubulin gene (*β-tub*), and partial calmodulin gene (*CAL*) (Appendix A) [26,27]. The PCR amplification conditions were established as follows: an initial denaturation step at 95 °C for 5 min, followed by 35 cycles of denaturation at 95 °C for 30 s, annealing temperature varied for the specific primers for 30 s (relative to the primers used), extension at 72 °C for 1 min, and a final extension step at 72 °C for 10 min [28,29,30]. The PCR products were analyzed via 1.5% agarose gel electrophoresis. Subsequently, the PCR products were submitted to Beijing Tsingke Biotech Co., Ltd. (Beijing, China) for DNA sequencing [31,32].

In this study, the ITS, *β-tub*, *EF1-α*, and *CAL* sequences of identified *Diaporthe* strains were obtained from GenBank and edited using Clustal X 1.83. The character weights were set to be equal during the editing process. Further, a phylogenetic tree of *Diaporthe* was developed via maximum likelihood using MEGA 7.0 according to these edited sequences.

### 2.3. Diaporthe Isolates Pathogenicity Testing

Founded on the identification results and Koch’s postulates, we performed pathogenicity tests on etiolated soybean seedlings using representative isolates of the identified *Diaporthe* species. Soybean seeds of equal size and number were placed 8 cm apart in pots containing moist vermiculite. A moist vermiculite layer was added, and the seeds were placed in cardboard boxes covered with black plastic bags for ten days to encourage etiolation. The etiolated soybean seedlings were placed in a shallow tray lined with three layers of moist paper. A 5-mm diameter plug of mycelium was inoculated into the center of the etiolated soybean seedlings. The tray was covered with a section of cling film to maintain optimum humidity. Following incubation at 25 °C for approximately 8 days, the diameter of the lesions formed was investigated and recorded.

### 2.4. Fungicides

The sensitivity of eight fungicides to representative *Diaporthe* isolates was evaluated. The fungicides included tebuconazole (96%; Shandong United Pesticide Industry Co. Ltd., Jinan, China), prothioconazole (98%; Shandong United Pesticide Industry Co. Ltd., Jinan, China), mefentrifluconazole (97%; Badische Anilin-und-Soda-Fabrik SE, Ludwigshafen, Germany), fludioxonil (98%; Shandong United Pesticide Industry Co. Ltd., Jinan, China), azoxystrobin (93%; Syngenta AG, Basel, Switzerland), phenamacril (95%; Jiangsu Pesticide Research Institute Co., Nanjing, China), pydiflumetofen (98%; Bayer AG, Syngenta Crop Protection Ltd., Leverkusen, Switzerland), and fluopyram (96%; Bayer AG, Leverkusen, Germany).

### 2.5. Sensitivity Assay of Diaporthe Isolates to Eight Fungicides

Sensitivity to fungicides has been typically determined in vitro by estimating the effective concentration required to lower radial mycelial growth by 50% (EC_50_) in fungicide-amended media relative to unamended media [33,34]. The efficacy of seven fungicides on the mycelial growth of 46 *Diaporthe* spp. on PDA was examined following the protocol outlined in our previous study, with the concentrations of the eight fungicides listed in Table 1 [35]. The median effective concentration (EC_50_) was computed using a linear model regression of the relative growth values (as a proportion of the control) against the log-transformed fungicide concentrations. Eight chemical fungicides were dissolved in dimethylsulfoxide (DMSO) to produce stock solutions with a concentration of 10^5^ μg/mL at 4 °C, and the final concentration of DMSO in the medium was 0.1% across all treatments. Mycelial plugs (5 mm in diameter) were removed from the edges of the colonies and transferred to the center of PDA plates with different concentrations of fungicide. Each treatment consisted of three replicates. The diameter of each colony was examined perpendicularly following ten days of incubation at 25 °C in the dark. The experiment was performed in duplicate [36].

### 2.6. Cloning of SdhA, B, C, and D Genes in D. longicolla

The DNA of *D. longicolla* was isolated from the mycelia grown on PDA media at 25 °C in the dark for 3 days using the cetyltrimethyl ammonium bromide (CTAB) procedure as outlined in Section 2.2. PCRs utilizing primers listed in Table 2, the reaction mixture was prepared to a final volume of 25 μL, containing 2 μL of fungal template DNA. PCR amplification of *SdhA*, *B*, *C*, and *D* contained 0.5 μL of each primer, 12.5 μL of 2 × GS Taq PCR Mix, and 9.5 μL of H_2_O. Amplification conditions were 95 °C for 3 min followed by 34 cycles of 95 °C for 30 s, 61 °C (*SdhA*), 63 °C (*SdhB*), 59 °C (*SdhC*), and 63 °C (*SdhD*) for 30 s and 72 °C for 1 min (2.5 min for *SdhA*) with a final DNA extension at 72 °C for 10 min. PCR products were sequenced by Beijing Tsingke Biotech Co., Ltd. (Beijing, China). Nucleotide sequences were assembled and aligned using Bio-Edit software version 7.0.8.1, and amino acid sequences were characterized following sequence analysis.

### 2.7. Analysis of SdhB and SdhC in D. longicolla

Mitochondrial complex II is normally composed of four subunits: SdhA, SdhB, SdhC, and SdhD subunits. According to previous studies, the major resistance mutation sites are located in the three subunits: SdhB, SdhC, and SdhD. For fluopyram and pydiflumetofen, the primary resistance mutation sites are situated in the SdhB and SdhC subunits. Therefore, this study aimed to clarify the sequence differences of the SdhB and SdhC subunits in seed-rot pathogens relative to other different species and produced a phylogenetic tree using MEGA version 7.0.26 software to illustrate this result.

### 2.8. Molecular Docking Analysis and Molecular Dynamics (MD) Simulations

Molecular docking was employed to investigate the binding model of target SDHI fungicides to Sdh proteins. The crystal structure of 3aeb, the Sdh protein from *Saccharomyces cerevisiae*, obtained from the Protein Data Bank, was used as a template for homology modeling. Sequence alignment of the Sdh proteins using DNAMAN revealed over 30% sequence identity with *D. longicolla* [37,38,39]. The Modeller v 9.19 program was used to perform homology modeling on the protein, in order to obtain a reasonable three-dimensional structural model of the target protein. The three-dimensional (3D) conformations of pydiflumetofen and fluopyram were optimized using an Amber14 force field, which is integrated within YASARA 5.0. The binding mode between the SDHI fungicides and Sdh proteins was studied using molecular docking in YASARA. The docking box size was set to 25 × 25 × 25 Å, with the default built-in ligand exhaustiveness and flexibility methods. All other parameters were kept at their default settings [40]. PyMOL 3.1.4.1 was used for visualization analysis. The optimal conformation was then selected for molecular dynamics simulation.

The Molecular dynamics simulations were conducted using AMBER 24 with the ff19SB force field for proteins, the GAFF2 force field for ligands, and the OPC water model for the ligand-protein complex system [41]. Complex systems were neutralized with Na^+^/Cl^−^ ions. The simulation workflow included energy minimization using the steepest descent and conjugate-gradient methods, followed by gradual heating from 0 K to 300 K and equilibration for 500 ps. A 100 ns production MD run was then performed. Trajectory analysis, including root mean square deviation (RMSD) and stable conformation extraction, was carried out using CPPTRAJ integrated into Amber 24 [42]. The equilibrated trajectory of each complex was sampled for MM-GBSA calculations using MMPBSA.py script integrated in AMBER 24 [43].

### 2.9. Synergistic Interaction of Mixed Fungicides

The sensitivity of the fungicide mixtures containing fludioxonil, mefentrifluconazole, and pydiflumetofen against *D. longicolla* was determined by the mycelial growth rate method as Section 2.5 [44]. The experiment integrated fludioxonil and mefentrifluconazole, fludioxonil, and pydiflumetofen with ratios of 1:0, 0:1, 1:1, 1:5, 5:1, and 10:1. Plugs of 5 mm diameter were obtained from the edge of the colony using a puncher and inoculated onto PDA medium containing various concentrations of the mixed fungicide concentration of mixed fungicide. The theoretical inhibition concentration of each combined fungicide (EC_50_ values) was determined by measuring the diameter of each colony in two perpendicular directions following five days of incubation at 25 °C in the dark, Values for the co-toxicity coefficient (CTC) were determined based on the EC_50_ values of each binary mixture using the following formula. [45].$$\mathrm{Toxicity}\;\mathrm{index}\;\left(\mathrm{TI}\right)\;\mathrm{of}\;A=100$$$$\mathrm{TI}\left(B\right)=\frac{{\mathrm{EC}}_{50\;}\left(\;A\;\right)}{{\;\mathrm{EC}}_{50}\left(B\right)}\times 100$$$$\mathrm{Actual}\;\mathrm{TI}\left(M\right)={\;\mathrm{EC}}_{50}(A)\times {\mathrm{EC}}_{50}(M)\times 100$$$$\mathrm{Theoretical}\;\mathrm{TI}\left(M\right)=\mathrm{TI}\left(A\right)\times \%\left(W\right)\;\mathrm{of}\;A\;\mathrm{in}\;M+\mathrm{TI}(B)\times \%\left(W\right)\;\mathrm{of}\;B\;\mathrm{in}\;M$$

A, B represent the individual fungicides used in this mixture; M indicates the mixture of two fungicides; W is the proportion of the fungicide in the mixture. The co-toxicity coefficient (CTC) is computed as the ratio between Actual TI (M) and Theoretical TI (M). If CTC ≥ 120, synergism is present. If 120 > CTC > 80, the interaction is additive. If CTC < 80, the interaction is antagonistic [46].

### 2.10. Statistical Analysis

Statistical analysis of the data was conducted using DPS software version 7.05. The data were presented as the mean ± standard error. Significant differences between means were identified using a multi-range least significant difference (LSD) test at a significance level of *p* = 0.05.

## 3. Results

### 3.1. D. longicolla Was Identified from a Diseased Soybean Stem

A total of 46 isolates were isolated from diseased soybeans in the Heilongjiang province of China, and these isolates developed colonies with white, compact aerial mycelium in the front with greenish-black stromata in the back of the colony in the Petri dish (Figure 1A) [47]. The spores of these species are ellipsoid or fusiform, transparent, and usually contain two oil globules (Figure 1B). The phylogenetic tree, developed using ITS, *EF1-α*, *CAL*, and *β-tub* gene sequences, supported the identification of the isolates as *D. longicolla*, consistent with the morphological results (Figure 1C). Additionally, through the pathogenicity of *Diaporthe* isolates, the results indicted all the *Diaporthe* isolates could infect etiolated seedlings of soybean, causing disease symptoms (Figure 1D). Combined with morphological, molecular identification and pathogenicity, all 46 *Diaporthe* isolates isolated from diseased soybean stems were confirmed to be *D. longicolla*.

### 3.2. Determination of EC_50_ Values of Eight Fungicides and Sensitivity Differentiation in Two SDHIs Against D. longicolla

Among the eight fungicides, except the two SDHIs, pydiflumetofen and fluopyram, fludioxonil, mefentrifluconazole, and azoxystrobin exhibited adequate inhibition of *D. longicolla*. *D. longicolla* was most sensitive to fludioxonil with an EC_50_ of about 0. 01 μg/mL, followed by mefentrifluconazole and azoxystrobin, whose EC_50_ values were below 1 μg/mL, but phenamacril was the least sensitive with an EC_50_ of 3.59 μg/mL among the six fungicides (Figure 2). And in this study, the sensitivity of *D. longicolla* to SDHI pydiflumetofen and SDHI fluopyram was determined, and it was found that pydiflumetofen and fluopyram belong to the SDHI fungicides, with a difference in sensitivity of *D. longicolla*. The SDHI pydiflumetofen exhibited high inhibitory activity against *D. longicolla* with mean EC_50_ values of 5.47 ± 0.63 μg/mL (Figure 2). In contrast, the SDHI fluopyram was insensitive, and its mean EC_50_ values of almost all isolates exceeded 100 μg/mL.

### 3.3. Characterization of the D. longicolla Sdh (A, B, C, and D) Genes

The four primer pairs Sdh-F/R produced four fragments, including the full-length *Sdh A*, *B*, *C*, and *D* genes and some flanking sequences. The *SdhA* gene was 2244 bp in length, encoding 741 amino acids, the *SdhB* gene encoded 281 amino acids at 873 bp in length, along with an intron 67 bp in length located at nucleotide position 407. The *SdhC* and *SdhD* genes were 552 bp and 576 bp in length, respectively, each encoding 181 acids. However, *SdhD* has an intron 50 bp in length located at nucleotide position 205.

### 3.4. Analysis Sdh (B and C) Proteins in D. longicolla

According to the significant sensitivity differences of *D. longicolla* to the two SDHI fungicides, we investigated and analyzed the domains of their SdhA, SdhB, SdhC, and SdhD protein. The SdhA protein of *D.longicolla* contains only the PTZ00139 domain (FAD_binding_2 domain), which is the succinate dehydrogenase [ubiquinone] flavoprotein subunit. For the SdhB protein, *D. longicolla* has only the PLN00129 superfamily domain which is the succinate dehydrogenase [ubiquinone] iron-sulfur subunit, and no transmembrane domain (http://smart.embl-heidelberg.de/) (accessed on 20 May 2024). The SdhC protein of *D. longicolla* contains only SQR domains, which belong to the succinate:quinone oxidoreductase (SQR) type C subfamily. And SQR catalyzes the oxidation of succinate to fumarate, coupled to the reduction of quinone to quinol, but it has three transmembrane domains. And for the SdhD protein, *D.longicolla* has the CybS domain, it encodes the small subunit (cybS) of cytochrome b in succinate-ubiquinone oxidoreductase (mitochondrial complex II) (Figure 3A). Additionally, phylogenetic trees demonstrated that the ShbA, SdhB, SdhC and SdhD subunits of *D. longicolla* were individually grouped in a single cluster, potentially accounting for the obvious difference in fungicide sensitivity of the two SDHIs, pydiflumetofen and fluopyram (Figure 3B).

### 3.5. Molecular Docking and MD Simulation of Two SDHI Fungicides with D. longicolla Sdh Proteins

Based on the sensitivity results of different species to the two SDHI fungicides, *D. longicolla* showed significant differences in sensitivity to fluopyram and pydiflumetofen. To explore the underlying mechanism, molecular docking and MD simulation were conducted. Initially, fluopyram and pydiflumetofen were individually docked into the binding pocket of Sdh proteins (Appendix A), and the docked complexes served as the initial conformations for MD simulations.

As shown in Figure 4A, the RMSD trends of fluopyram and pydiflumetofen began to converge after 40 ns. The ligand RMSD of pydiflumetofen stabilized at 2.25 Å approximately, fluctuations within 1 Å. However, fluopyram stabilized at around 2.00 Å, fluctuations within 2 Å, indicated that pydiflumetofen bound more stably to the Sdh proteins than fluopyram. Subsequently, representative protein−ligand conformations and the binding free energy (Figure 4B,C) were analyzed from the final 1 ns of the simulations. The binding free energy (ΔG_bind_) of fluopyram with Sdh proteins was −23.92 kcal/mol, higher than that of pydiflumetofen with the Sdh proteins (−30.02 kcal/mol). This difference may explain why pydiflumetofen exhibits stronger binding with the Sdh proteins.

Further analysis of the electrostatic potential and structural conformations of the binding pocket revealed that it was predominantly hydrophobic. Both fluopyram and pydiflumetofen formed hydrophobic interactions with surrounding residues. Fluopyram formed hydrophobic bonds with the residues C_His146, C_Gly143, C_Val144, C_Leu175, D_-Val30 and D_Ala26 of Sdh proteins. Pydiflumetofen formed hydrophobic interactions with the residues C_His146, C_His139, C_Trp46 D_-Val30, D_Ala27 and D_Ala26 of the Sdh proteins. Pydifluorofen was positioned closer to the D subunit and formed more hydrophobic interactions, which may contribute to its differing activity compared to fluopyram. However, neither ligand exhibited strong electrostatic interactions, such as hydrogen bonds or π-π stacking. These findings suggested that structural modifications of fluopyram and pydiflumetofen could introduce hydrogen bond donor fragments, enabling hydrogen bonding interactions with the oxygen acceptor atoms of Gly143 and Ala27.

### 3.6. Synergy of Fludioxonil, Mefentrifluconazole, and Pydiflumetofen Against D. longicolla

Based on the sensitivity results of eight fungicides again *D. longicolla*, it was found that fludioxonil and mefentrifluconazole had the best inhibitory effect, and SDHI fungicide pydiflumetofen also exhibited good inhibition again *D. longicolla*. To investigate whether there is a synergistic effect of the three fungicides, fludioxonil and mefentrifluconazole, fludioxonil and pydiflumetofen were mixed as the binary mixture and tested the sensitivities against *D. longicolla*. It showed that the mixture of fludioxonil and mefentrifluconazole, fludioxonil and pydiflumetofen, showed synergism efficiency only at the ratio of 1:5, and the CTC values were 126.39 and 140.00 respectively (Table 3).

## 4. Discussion

Soybeans are threatened by many *Diaporthe* species, causing soybean stem blight that negatively affects yield and quality, and pathogens and dominant species vary by distinct region [48,49,50,51]. Among them, the pathogen *D. longicolla* was reported causing soybean stem blight [52]. In this study, integrating morphology and molecular identification, we obtained 46 *D. longicolla,* consistent with the previous study, but the number of isolates of *D. longicolla* was 25% percent less than those isolated from diseased soybean plants [17]. *D. longicolla* has a wide range of hosts, which can infect capsicum, tomato, and soybeans [53]. While it was isolated in relatively low levels, it can still infect many plants and can aggravate soybean diseases through synergistic infection with *Fusarium* species and other pathogens, so the sensitivity and infection mechanisms of *D. longicolla* bear investigation.

In this study, the fungicides tebuconazole, mefentrifluconazole, and fludioxonil demonstrated adequate inhibition of *D. longicolla,* while in our previous study, these fungicides demonstrated similar inhibition for *Fusarium* species. These factors should, therefore, be considered in the field for controlling soybean root rot and soybean stem blight caused by various pathogens, and in practice, the alternate rotation of different mechanism fungicides should also be taken into consideration and could be matched with fungicides that could control oomycetes or inhibit various fungi simultaneously. There is some interest in the sensitivity result for SDHI fluopyram, which showed little or no inhibition of both *D. longicolla*, while SDHI pydiflumetofen exhibited good inhibition. Therefore, prior to using fungicides with the same mode of action, the differences between pathogens in plant diseases should be carefully considered, and the sensitivity of different *D. longicolla* to different SDHIs requires further investigation in the future.

To further document the mechanisms underlying the difference in sensitivity between two SDHI fungicides: fluopyram and pydiflumetofen, we firstly investigated the binding interactions of *D. longicolla* Sdh proteins to SDHIs based on molecular docking [54,55]. The highest-scoring docking was the complex, having one of the top-ranked poses [56]. Molecular docking results revealed that pydiflumetofen exhibits superior binding with the Sdh proteins, and it indicated that the SdhC subunit plays an important role in this inteaciton. In *F. pseudograminearum*, the recent research also demonstrated that the sensitivity of pydiflumetofen is closely associated with SdhC_1_ subunit, with point mutations FpSdhC_1_^A83V^ or FpSdhC_1_^R86K^ conferring pydiflumetofen resistance [57]. In the study by Chen et al., single amino acid mutations in FaSdhC_1_ confer pydiflumetofen resistance in *F. asiaticum* [22]. Combining the docking result and previous study showing that SDHI pydiflumetofen was mainly bound to SdhC protein.

Further molecular dynamics simulations of fluopyram and pydiflumetofen revealed that both bind to the Sdh proteins through hydrophobic interactions. However, fluopyram specifically forms hydrophobic bonds with the C_Gly143, C_Val144, and C_Leu175 residues of the Sdh protein, pydiflumetofen specifically forms hydrophobic bonds with the C_His139, C_Trp46, and D_Ala27 residues of the Sdh proteins, which leads to different binding activities with the Sdh proteins. Additionally, pydiflumetofen could also interact with amino acid residues of the SdhD protein, suggested that the SdhC protein and SdhD proteins synergistically bind with the fungicide, resulting in a stronger binding of pydiflumetofen to the Sdh proteins.

An interesting phenomenon was observed in this study: SDHI pydiflumetofen and fluopyram have different inhibitory effects on *D. longicolla*. Despite the relatively poor activity of fluopyram, molecular docking and molecular dynamics studies revealed that that SDHI fluopyram also has several binding sites with the SdhC and SdhD proteins This indicated that the molecular docking and molecular dynamics results were only an auxiliary validation and the actual binding sites of pathogens to distinct fungicides still needed to be practiced in vivo or ex vivo experiments and the only by these results, it cannot fully explain the actual sensitivities of SDHI fungicides against *D. longicolla* [58].

In conclusion, 46 *Diaporthe* isolates were obtained and identified as *D. longicolla*, causing soybean stem blight in the Heilongjiang province of China from 2021 and 2022. Fludioxonil, mefentrifluconazole, and tebuconazole exhibited an adequate level of inhibition of *D. longicolla*, and could be recommended for the control of soybean stem blight in the field. The binding capacity of two SDHIs, pydiflumetofen and fluopyram, to Sdh proteins varied, further elucidating their different inhibitory effects against *D. longicolla*. However, according to the sensitivity results of mixed fungicides, fludioxonil, mefentrifluconazole and pydiflumetofen against *D. longicolla*, most combinations exhibited antagonistic effect, while synergistic efficiency was observed only at a ratio of 1:5. Therefore, further practice and exploration of the actual efficacy of mixed fungicides in the field are needed.

## Figures and Tables

**Figure 1 jof-11-00292-f001:**
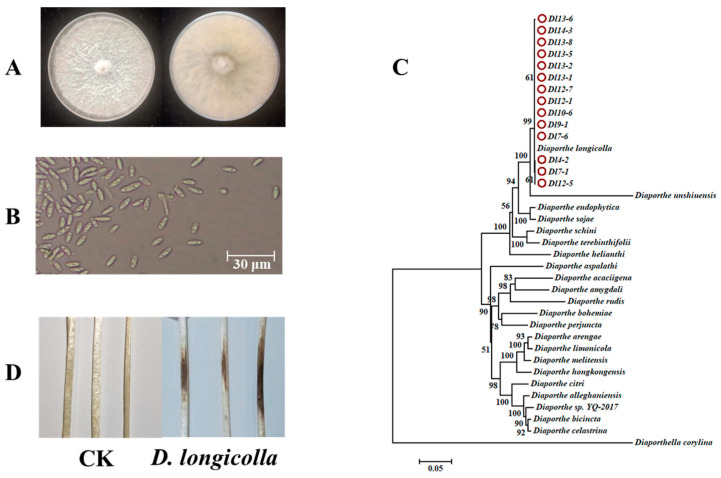
Identification of *Diaporthe longicolla* obtained from soybean rots according to morphological and phylogenesis analysis in Heilongjiang, China. (**A**) Typical colonies of *D. longicolla* observed after culturing for 3 days on PDA medium; (**B**) Macroconidia morphology of *D. longicolla* following incubation for 14 days on WA medium; (**C**) The *ITS*, *β-tub*, *EF1-α*, and *CAL* sequences of *D. longicolla* isolates were obtained from GenBank, and the phylogenetic tree was developed via maximum likelihood using MEGA 7.0 based on these three sequences; (**D**) Pathogenicity of *D. longicolla* on etiolated soybean seedlings inoculated in the dark at 25 °C. CK represented the control inoculated with an empty PDA medium of 5 mm diameter, and the lesion diameter was determined after 7 days.

**Figure 2 jof-11-00292-f002:**
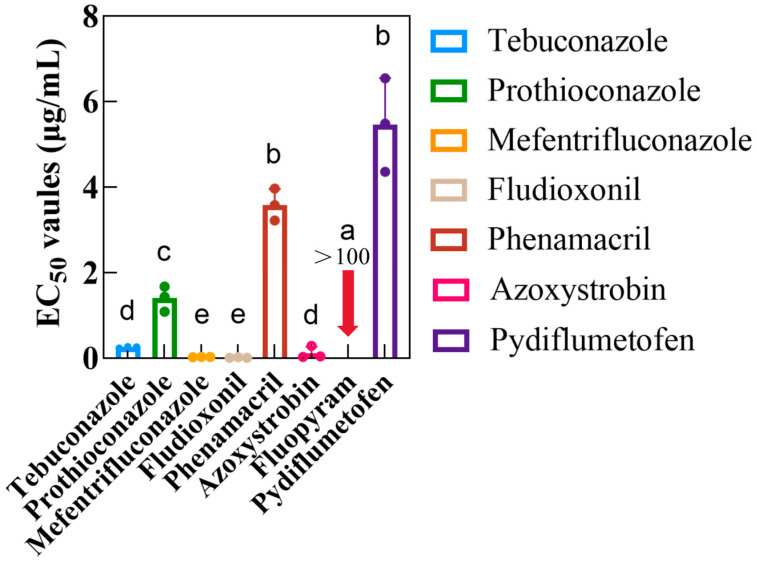
Bar chart of sensitivity with *Diaporthe longicolla* to eight fungicides. Eight fungicides in total, including tebuconazole, prothioconazole, mefentrifluconazole, fludioxonil, phenamacril, azoxystrobin, pydiflumetofen and fluopyram were assessed. Each treatment consisted of three replicates. The diameter of each colony was examined perpendicularly following five days of incubation at 25 °C in the dark. The experiment was performed in duplicate. Each column represents the sensitivity of three strains to a fungicide with three independent biological replicates, with error bars representing three biological replicates of the three strains. Different letters represent significant differences detected by one-way ANOVA (*p* < 0.05).

**Figure 3 jof-11-00292-f003:**
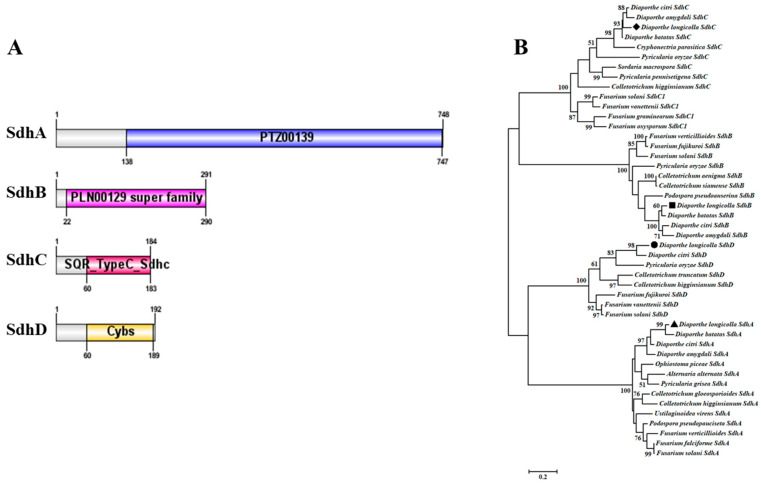
Domain structure and phylogenetic tree examination of *Diaporthe longicolla* SdhA, SdhB, SdhC and SdhD proteins. (**A**) The structural domains of the SdhA, SdhB, SdhC and SdhD subunits in *D. longicolla*. The protein domains were mapped using DOG 2.0 software, with different colors representing different domains on the corresponding subunits; (**B**) The SdhA, SdhB, SdhC and SdhD sequences of *D. longicolla* isolates were obtained from GenBank, and the phylogenetic tree were developed via maximum likelihood using MEGA 7.0 according to these sequences in *D. longicolla*. Black triangles represents SdhA protein, black squares represents SdhB protein, black diamonds represents SdhC protein, and black circles represents SdhD protein.

**Figure 4 jof-11-00292-f004:**
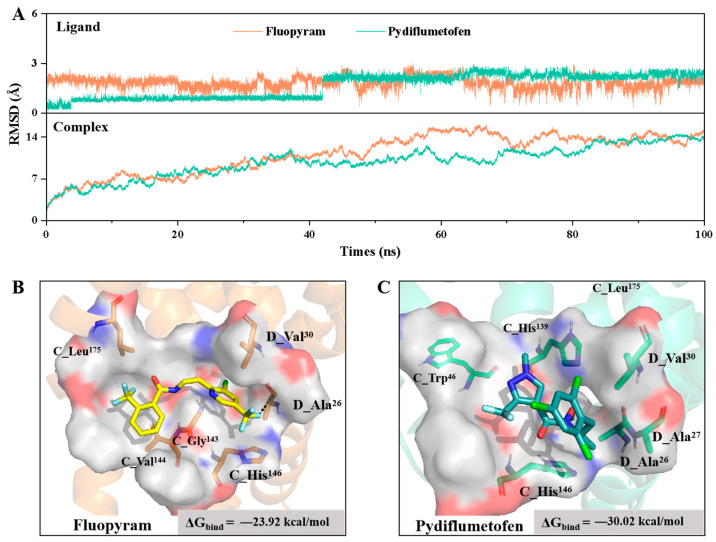
Molecular dynamics simulation of SDHI fungicides with the *Diaporthe longicolla* Sdh proteins. (**A**) The average RMSD of the fluopyram and pydiflumetofen system; (**B**) Binding mode of fluopyram. Fluopyram formed hydrophobic bonds with the residues in black font, and the binding free energy was −23.92 kcal/mol; (**C**) Pydiflumetofenformed hydrophobic bonds with the residues in black font, and the binding free energy was −30.02 kcal/mol.

**Table 1 jof-11-00292-t001:** Fungicides and concentration gradients of eight fungicides acting on *Diaporthe longicolla*.

Fungicide	Concentration Gradient (μg/mL)
Tebuconazole	0, 0.1, 0.4, 0.8, 1, 2.5
Prothioconazole	0, 1,5, 10, 25, 50, 100
Mefentrifluconazole	0, 0.005, 0.05, 0.25, 1, 5, 20
Fludioxonil	0, 0.01, 0.02, 0.05, 0.1, 0.25, 1
Phenamacril	0, 0.05, 0.25, 0.5, 1.25, 2.5, 10
Pydiflumetofen	0, 0.2, 0.5, 1, 2.5, 5, 10
Fluopyram	0, 1, 10, 20, 50, 100
Azoxystrobin	0, 0.01, 0.02, 0.05, 0.1, 0.25, 1

**Table 2 jof-11-00292-t002:** Details of primers employed to amplify the *Sdh* (*A*, *B*, *C*, and *D*) genes isolates *Diaporthe longicolla* isolates.

Primer	Primer Sequence (5′–3′)	Product Length (bp)
*SdhA*(F)	TTATCACGGCTGATGTCGGATCG	2000
*SdhA*(R)	CCTTCTGGAAAGACAAGGTGTGCTT	
*SdhB*(F)	ATGGCTGCTCTCCGCTCTACCTCCA	850
*SdhB*(R)	TACACAGCCATCTCCTTCTTGATCT	
*SdhC*(F)	ATGATTACCTCAAGGGCAGG	550
*SdhC*(R)	TTACAGCAGGAACGCAACACCCA	
*SdhD*(F)	ATGGCATCCACTGCTCGTTCGGCT	600
*SdhD*(R)	TCATGCTCGCCAAAGCCTCTT	

**Table 3 jof-11-00292-t003:** The synergistic effect of fludioxonil and mefentrifluconazole, fludioxonil and pydiflumetofen against against *Diaporthe longicalla* at different ratios.

Mixture (A:B)	Mixture Ratios (A:B)	EC_50_ (μg/mL)	Co-Toxicity Index	Interaction Efficiency
Mefentrifluconazole and fludioxonil	1:0	0.0663	-	-
	0:1	0.0047	-	-
	1:1	0.5070	17.3131	antagonistic
	1:5	0.0044	126.3899	synergism
	5:1	0.0377	55.2262	antagonistic
Pydiflumetofen and fludioxonil	1:0	6.9921	-	-
	0:1	0.0057	-	-
	10:1	0.4220	1.4857	antagonistic
	1:5	0.0049	695.1258	synergism
	5:1	0.2379	2.8747	antagonistic

A represents the mixed fungicide mefentrifluconazole or pydiflumetofen, and B represents fludioxonil.

## Data Availability

Data will be made available on request.

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
