# Peer review of "Exploring Fungicide Sensitivity in Soybean Stem Blight Pathogen Diaporthe longicolla, Emphasizing Genetic Variability Impact on Response to SDHI Fungicides Fluopyram and Pydiflumetofen"

_jof, 2025, doi:10.3390/jof11040292_

Round 1

Reviewer 1 Report

The study provides important information about a pathogen associated with Soybean Stem Blight in China. It is concise, well written and small notes are made.

Abstract

  • There is no need to cite anamorph and teleomorph (one fungus one name)

Methods

  • Line 96: they are not specific primers
  • Line 108: The ideal would be a more robust phylogeny as Bayesian inference
  • Line 217-225: The caption was too short for the four figures, for example, the phylogeny was not well explained, the statistical method could have been included, which is the bar
  • Line 250: change am to an

References

  • Check scientific name without italics

Author Response

Thanks very much for your professional comments on our JOF manuscript (jof-3537544), originally entitled "Exploring Fungicide Sensitivity in Soybean Stem Blight Pathogen Diaporthe longicolla, Emphasizing Genetic Variability Impact on Response to SDHI Fungicides Fluopyram and Pydiflumetofen". We have revised the manuscript according to the comments of you, and all new modifications have been noted in the tracked manuscript. Meanwhile, a clean copy version was attached. Responses were made to each point as follows:

Reviewer #1:

Comments 1: The study provides important information about a pathogen associated with Soybean Stem Blight in China. It is concise, well written and small notes are made.

Response 1: Thanks for your kindly comments.

Comments 2: Abstract: There is no need to cite anamorph and teleomorph (one fungus one name).

Response 2: The anamorph name have been deleted in the revised manuscript.

Comments 3: Methods: Line 96: they are not specific primers.

Response 3: The word ‘specific’ have been deleted in the revised manuscript.

Comments 4: Methods: Line 108: The ideal would be a more robust phylogeny as Bayesian inference.

Response 4: Thanks for your kindly suggestion. There is no Bayesian inference algorithm in MEGA 7.0. According to other reports, the phylogenetic tree of Diaporthe was also developed using maximum likelihood in MEGA 7.0, and it can also reflect an accurate evolutionary relationship. 

Sun, W., Huang, S., Xia, J., Zhang, X., Li, Z. (2021). Morphological and molecular identification of Diaporthe species in south-western China, with description of eight new species. MycoKeys. 77:65-95. https://doi.77:65-95. 10.3897/mycokeys.77.59852.

Sun, X., Cai, X., Pang, Q., Zhou, M., Zhang, W., Chen, Y., Bian, Q. (2021). First report of Diaporthe longicolla causing leaf spot on Kalanchoe pinnata in China. Plant Dis. https://doi.10.1094/PDIS-12-20-2681-PDN.  

Comments 5: Methods: Line 217-225: The caption was too short for the four figures, for example, the phylogeny was not well explained, the statistical method could have been included, which is the bar.

Response 5: Thanks for your kindly suggestion. The captions of the four figures have been modified to make them clearer in the revised manuscript.

Comments 6: Methods: Line 250: change am to an.

Response 6: The ‘am’ has been changed to ‘an’ in the revised manuscript.

Comments 7: References: Check scientific name without italics.

Response 7: The scientific name without italics have been checked and modified in the revised manuscript.

Reviewer 2 Report

The following are suggestions that authors should consider to improve the technical and scientific quality of the manuscript:
-The algorithm or tool used for sequence alignment should be specified.
Although an Amber14 force field was mentioned as being used for conformation optimization, it would be helpful to specify if any additional methods, such as energy minimization or molecular dynamics methodology, were used.
-Details on the parameters used in molecular docking, such as search box size, ligand exhaustiveness, and flexibility methods, should be included.
-The purpose of the MOPAC optimization should be briefly explained. Were the configurations obtained from docking subsequently optimized using MOPAC software to improve the accuracy of interaction energies and ligand positions?
-A description of the metrics used to evaluate the docking results, such as binding energy, RMSD values, and key ligand-protein interactions, could be added.
The analysis and discussion of results is very brief. The following suggestions may be considered:
-Given that D. longicolla's sensitivity to fungicides varies, it would be helpful to investigate further the differences in interactions between the two fungicides (pydiflumetofen and fluopyram) and the D. longicolla SdhC proteins. Are there structural features in the D. longicolla proteins that could influence the difference in docking scores?
-Docking scores can be associated with binding energies. Explain how the scores (5.35 vs. 4.53) correlate with binding energies and what implications this has for the stability of fungicide binding.
-Interactions at the atomic level that could influence the different docking scores should be examined. Which residues in the SdhC proteins are involved in the interactions with pydiflumetofen and fluopyram? Analyze whether these interactions are key to binding affinity and whether they are specific to each fungicide.
- It should be considered whether the binding of fungicides induces any significant conformational changes in the protein. Molecular dynamics simulations could be helpful to verify whether these differences in docking scores are due to a conformational change that affects the accessibility of the binding site. They also allow for the assessment of the stability of the complexes and the flexibility of the residues in the binding site to provide insight into the temporal stability of interactions between fungicides and proteins, which is crucial for understanding the potency and durability of inhibition.
- It should be explored how the structural characteristics of fungicides (e.g., polarity, size, and shape) may influence their ability to fit into the SdhC protein binding site. Are there chemical modifications to the structure that could improve binding affinity or selectivity?
-Authors should review and modify the text between lines 30-36, 42-45, 73-76, 91-95, 100-104, 111-119, 122-127, 135-144, 147-157, 170-176, and
197-202, to reduce overlap with literature sources.
-Regarding the references cited, 13 of the 55 are outside a 5-year time window, which I consider long-lived. It would be important to update these references. Specifically, review the relevance of the following references: 1, 2, 4, 5, 8, 11, 15, 16, 17, 28, 36, 41, 42.

The following are suggestions that authors should consider to improve the technical and scientific quality of the manuscript:
-The algorithm or tool used for sequence alignment should be specified.
Although an Amber14 force field was mentioned as being used for conformation optimization, it would be helpful to specify if any additional methods, such as energy minimization or molecular dynamics methodology, were used.
-Details on the parameters used in molecular docking, such as search box size, ligand exhaustiveness, and flexibility methods, should be included.
-The purpose of the MOPAC optimization should be briefly explained. Were the configurations obtained from docking subsequently optimized using MOPAC software to improve the accuracy of interaction energies and ligand positions?
-A description of the metrics used to evaluate the docking results, such as binding energy, RMSD values, and key ligand-protein interactions, could be added.
The analysis and discussion of results is very brief. The following suggestions may be considered:
-Given that D. longicolla's sensitivity to fungicides varies, it would be helpful to investigate further the differences in interactions between the two fungicides (pydiflumetofen and fluopyram) and the D. longicolla SdhC proteins. Are there structural features in the D. longicolla proteins that could influence the difference in docking scores?
-Docking scores can be associated with binding energies. Explain how the scores (5.35 vs. 4.53) correlate with binding energies and what implications this has for the stability of fungicide binding.
-Interactions at the atomic level that could influence the different docking scores should be examined. Which residues in the SdhC proteins are involved in the interactions with pydiflumetofen and fluopyram? Analyze whether these interactions are key to binding affinity and whether they are specific to each fungicide.
- It should be considered whether the binding of fungicides induces any significant conformational changes in the protein. Molecular dynamics simulations could be helpful to verify whether these differences in docking scores are due to a conformational change that affects the accessibility of the binding site. They also allow for the assessment of the stability of the complexes and the flexibility of the residues in the binding site to provide insight into the temporal stability of interactions between fungicides and proteins, which is crucial for understanding the potency and durability of inhibition.
- It should be explored how the structural characteristics of fungicides (e.g., polarity, size, and shape) may influence their ability to fit into the SdhC protein binding site. Are there chemical modifications to the structure that could improve binding affinity or selectivity?
-Authors should review and modify the text between lines 30-36, 42-45, 73-76, 91-95, 100-104, 111-119, 122-127, 135-144, 147-157, 170-176, and
197-202, to reduce overlap with literature sources.
-Regarding the references cited, 13 of the 55 are outside a 5-year time window, which I consider long-lived. It would be important to update these references. Specifically, review the relevance of the following references: 1, 2, 4, 5, 8, 11, 15, 16, 17, 28, 36, 41, 42.

Author Response

Thanks very much for your professional comments on our JOF manuscript (jof-3537544), originally entitled "Exploring Fungicide Sensitivity in Soybean Stem Blight Pathogen Diaporthe longicolla, Emphasizing Genetic Variability Impact on Response to SDHI Fungicides Fluopyram and Pydiflumetofen". We have revised the manuscript according to the comments of you, and all new modifications have been noted in the tracked manuscript. Meanwhile, a clean copy version was attached. Responses were made to each point as follows:

Reviewer #2:

Comments 1: The algorithm or tool used for sequence alignment should be specified. Although an Amber14 force field was mentioned as being used for conformation optimization, it would be helpful to specify if any additional methods, such as energy minimization or molecular dynamics methodology, were used.

Response 1: Thanks for your kindly suggestion. We have modified in the revised manuscript. 

  1. The DNAMAN was used for sequence alignment.
  2. The crystal structure of 3aeb, the Sdhproteinsfrom Saccharomyces cerevisiae, were obtained from the Protein Data Bank, were used as a template for homology modeling. Sequence alignment of the Sdh proteins by DNAMAN unveiled over 30% sequence identity in D. longicolla. The Modeller v 9.19 program was used to perform homology modeling on the protein, in order to obtain a reasonable three-dimensional structural model of the target protein. The three-dimensional (3D) conformations of pydiflumetofen and fluopyram were optimized using an Amber14 force field, which is integrated within YASARA. 

Comments 2: Details on the parameters used in molecular docking, such as search box size, ligand exhaustiveness, and flexibility methods, should be included.

Response 2: Thanks for your kindly suggestion. We have modified in the revised manuscript. The binding mode between the SDHI fungicides and Sdh proteins were studied using molecular docking in YASARA. The docking box size was set to 25×25×25Å, with the default built-in ligand exhaustiveness and flexibility methods. All other parameters were kept at their default setting.

Comments 3: The purpose of the MOPAC optimization should be briefly explained. Were the configurations obtained from docking subsequently optimized using MOPAC software to improve the accuracy of interaction energies and ligand positions?

Response 3: Thanks for your kindly suggestion. The description of MOPAC in the last manuscript was not accurate in the last manuscript. The molecular docking between SDHI fungicides and Sdh proteins was conducted by YASARA, then visualized by PyMOL. The optimal conformation was selected for molecular dynamics simulation by AMBER 24. This description has been modified in the revised manuscript.

Comments 4: A description of the metrics used to evaluate the docking results, such as binding energy, RMSD values, and key ligand-protein interactions, could be added.

Response 4: Thanks for your kindly suggestion. Based on your suggestion, we have conducted further molecular dynamics simulations of the SDHI fungicides with the Sdh proteins, and analyzed the corresponding results, including binding energy, RMSD values, and key ligand-protein interactions. The detailed findings have been incorporated into the revised manuscript. Additionally, as the current molecular dynamics results indicated that ligands primarily interact with proteins through hydrophobic interactions, polar interactions are not reflected in the figure. Therefore, an electrostatic potential surface map has been added to illustrate the stable binding mode between ligands and protein.

Comments 5: Given that D. longicolla's sensitivity to fungicides varies, it would be helpful to investigate further the differences in interactions between the two fungicides (pydiflumetofen and fluopyram) and the D. longicolla SdhC proteins. Are there structural features in the D. longicolla proteins that could influence the difference in docking scores?

Response 5: Thanks for your kindly suggestion. Based on your suggestion, the molecular dynamics results have been added to the revised manuscript. The results indicated that fluopyram and pydiflumetofen ligands primarily interacted with Sdh proteins through hydrophobic interactions, which ultimately lead to the observed differences in activity. The polar interactions are not found in the both ligands.

Comments 6: Docking scores can be associated with binding energies. Explain how the scores (5.35 vs. 4.53) correlate with binding energies and what implications this has for the stability of fungicide binding.

Response 6: The docking score in last manuscript is the binding energy, which reflects the stability of the binding between the fungicides and the protein. The figure of the molecular docking has been placed in the supplementary materials.

Based on your suggestion, we analyzed the binding free energy after molecular dynamics simulations and found it to be consistent with the experimental results. The corresponding content has been added to the revised manuscript. “The binding free energy (ΔGbind) of fluopyram with the Sdh proteins was -23.92 kcal/mol, higher than that of pydiflumetofen with the Sdh proteins (-30.02 kcal/mol). This difference may explain why pydiflumetofen exhibits stronger binding to the Sdh proteins”.

Comments 7: Interactions at the atomic level that could influence the different docking scores should be examined. Which residues in the SdhC proteins are involved in the interactions with pydiflumetofen and fluopyram? Analyze whether these interactions are key to binding affinity and whether they are specific to each fungicide.

Response 7: Thanks for your kindly suggestion. This part has been comprehensively described in the results section of the revised manuscript

Fluopyram formed hydrophobic bonds with the residues C_His146, C_Gly143, C_Val144, C_Leu175, D_-Val30 and D_Ala26 of SdhC protein. Pydiflumetofen formed hydrophobic bonds with the residues C_His146, C_His139, C_Trp46 D_-Val30, D_Ala27 and D_Ala26 of SdhC protein. Pydifluorofen was positioned closer to the D subunit and formed more hydrophobic interactions, which may contribute to its differing activity compared to fluopyram. 

Comments 8: It should be considered whether the binding of fungicides induces any significant conformational changes in the protein. Molecular dynamics simulations could be helpful to verify whether these differences in docking scores are due to a conformational change that affects the accessibility of the binding site. They also allow for the assessment of the stability of the complexes and the flexibility of the residues in the binding site to provide insight into the temporal stability of interactions between fungicides and proteins, which is crucial for understanding the potency and durability of inhibition.

Response 8: Based on your suggestions, we performed molecular dynamics simulations on the complexes of pydiflumetofen and fluopyram with the Sdh proteins. The specific structures are as follows:

Molecular docking revealed that pydiflumetofen and fluopyram were respectively docked to the binding pocket of Sdh proteins, and the docked complexes were used as the initial conformation for MD simulation. The MD results indicated that the RMSD trends of fluopyram and pydiflumetofen began to converge after 40 ns (Fig. 4A). The ligand RMSD of pydiflumetofen stabilized at approximately 2.25 Å, with fluctuations within 1 Å. However, fluopyram stabilized at around 2.00 Å, fluctuations within 2 Å, indicated that pydiflumetofen bound more stably to the Sdh proteins than fluopyram. Subsequently, representative protein-ligand conformations and the binding free energy (Fig. 4B and 4C) were analyzed from the final 1 ns of the simulations. The binding free energy (ΔGbind) of fluopyram with the Sdh proteins was -23.92 kcal/mol, higher than that of pydiflumetofen with the Sdh proteins (-30.02 kcal/mol). This difference may explain why pydiflumetofen exhibits stronger binding to the Sdh proteins.

Comments 9: It should be explored how the structural characteristics of fungicides (e.g., polarity, size, and shape) may influence their ability to fit into the SdhC protein binding site. Are there chemical modifications to the structure that could improve binding affinity or selectivity

Response 9: Based on your suggestions, we analyzed the binding characteristics between the fungicides structures and Sdh proteins using the electrostatic potential surface and key amino acid interactions of stable conformations derived from molecular dynamics simulation results. Furthermore, we proposed corresponding structural modification strategies to enhance binding affinity, providing guidance for future optimization. The details are as follows:

Further analysis of the electrostatic potential and structural conformations of the binding pocket revealed that was predominantly hydrophobic. Both fluopyram and pydiflumetofen formed hydrophobic interactions with surrounding residues. Fluopyram formed hydrophobic bonds with the residues C_His146, C_Gly143, C_Val144, C_Leu175, D_-Val30 and D_Ala26 of SdhC protein. Pydiflumetofen formed hydrophobic bonds with the residues C_His146, C_His139, C_Trp46 D_-Val30, D_Ala27 and D_Ala26 of SdhC protein. Pydifluorofen was positioned closer to the D subunit and formed more hydrophobic interactions with, which may contribute to its differing activity compared to fluopyram. However, neither ligand exhibited strong electrostatic interactions, such as hydrogen bonds or π-π stacking. These findings suggested that structural modifications of fluopyram and pydiflumetofen could introduce hydrogen bond donor fragments, enabling hydrogen bonding interactions with the oxygen acceptor atoms of Gly143 and Ala27.

Comments 10: Authors should review and modify the text between lines 30-36, 42-45, 73-76, 91-95, 100-104, 111-119, 122-127, 135-144, 147-157, 170-176, and 197-202, to reduce overlap with literature sources.

Response 10: The references you mentioned have been modified to reduce overlap with literature sources in the revised manuscript.

Comments 11: Regarding the references cited, 13 of the 55 are outside a 5-year time window, which I consider long-lived. It would be important to update these references. Specifically, review the relevance of the following references: 1, 2, 4, 5, 8, 11, 15, 16, 17, 28, 36, 41, 42.

Response 11: The long-lived references have been checked and modified in the revised manuscript.

Reviewer 3 Report

Overall, the work is well-structured, and this contribution should be considered for publication after addressing the following comments.

  1. The abstract is very well written, but needs a small adjustment, if possible. However, I would like to know from the authors whether these values of losses related to the pathogen are measured in the literature, since it was mentioned that it is a very important pathosystem in soybeans. If you have the data on these losses, I suggest including them, for a better understanding of the problem to be controlled.
  2. The introduction is well written. The authors did a good review, in a more concise way, but covering all the important points for a good understanding of this research.
  3. Materials and Methods - Although 3 replicates are a statistically standard value, and their repetition is even more so, I suggest that the authors, in other works, include a greater number of replicates in each treatment. Sometimes some mycelial growth of fungi presents very inconclusive data, or does not grow well on plates. However, by working with such a small number of replicates, they end up having to repeat that treatment.
  4. Results - The results are interesting, explaining key points related to the research data. The graphs and figures are well constructed and easy to understand. Classic work in applied plant pathology. Very well written, and the data presented.
  5. Conclusions - In their conclusions, the authors were happy to state that these data need to be proven in the field, since laboratory data are not always reproduced in the field due to the large number of variables involved, both climatic and varietal, etc. The conclusion of the work was good.

Overall, the work is well-structured, and this contribution should be considered for publication after addressing the following comments.

  1. The abstract is very well written, but needs a small adjustment, if possible. However, I would like to know from the authors whether these values of losses related to the pathogen are measured in the literature, since it was mentioned that it is a very important pathosystem in soybeans. If you have the data on these losses, I suggest including them, for a better understanding of the problem to be controlled.
  2. The introduction is well written. The authors did a good review, in a more concise way, but covering all the important points for a good understanding of this research.
  3. Materials and Methods - Although 3 replicates are a statistically standard value, and their repetition is even more so, I suggest that the authors, in other works, include a greater number of replicates in each treatment. Sometimes some mycelial growth of fungi presents very inconclusive data, or does not grow well on plates. However, by working with such a small number of replicates, they end up having to repeat that treatment.
  4. Results - The results are interesting, explaining key points related to the research data. The graphs and figures are well constructed and easy to understand. Classic work in applied plant pathology. Very well written, and the data presented.
  5. Conclusions - In their conclusions, the authors were happy to state that these data need to be proven in the field, since laboratory data are not always reproduced in the field due to the large number of variables involved, both climatic and varietal, etc. The conclusion of the work was good.

Author Response

Thanks very much for your professional comments on our JOF manuscript (jof-3537544), originally entitled "Exploring Fungicide Sensitivity in Soybean Stem Blight Pathogen Diaporthe longicolla, Emphasizing Genetic Variability Impact on Response to SDHI Fungicides Fluopyram and Pydiflumetofen". We have revised the manuscript according to the comments of you, and all new modifications have been noted in the tracked manuscript. Meanwhile, a clean copy version was attached. Responses were made to each point as follows:

Reviewer #3:

Overall, the work is well-structured, and this contribution should be considered for publication after addressing the following comments.

Comments 1: The abstract is very well written, but needs a small adjustment, if possible. However, I would like to know from the authors whether these values of losses related to the pathogen are measured in the literature, since it was mentioned that it is a very important pathosystem in soybeans. If you have the data on these losses, I suggest including them, for a better understanding of the problem to be controlled.

Response 1: Thanks for your kindly suggestion.

Based on the current field disease situation, this disease is expected to cause a 30% reduction in soybean yield. However, further investigation is needed, so it was not included in the abstract. We will continue to investigate its impact and provide further reports in the future.

Comments 2: The introduction is well written. The authors did a good review, in a more concise way, but covering all the important points for a good understanding of this research.

Response 2: Thanks for your kindly comment.

Comments 3: Materials and Methods - Although 3 replicates are a statistically standard value, and their repetition is even more so, I suggest that the authors, in other works, include a greater number of replicates in each treatment. Sometimes some mycelial growth of fungi presents very inconclusive data, or does not grow well on plates. However, by working with such a small number of replicates, they end up having to repeat that treatment.

Response 3: Thanks for your kindly suggestion. We will include a greater number of replicates in each treatment in our other works.

Comments 4: Results - The results are interesting, explaining key points related to the research data. The graphs and figures are well constructed and easy to understand. Classic work in applied plant pathology. Very well written, and the data presented.

Response 4: Thanks for your kindly comment.

Comments 5: Conclusions - In their conclusions, the authors were happy to state that these data need to be proven in the field, since laboratory data are not always reproduced in the field due to the large number of variables involved, both climatic and varietal, etc. The conclusion of the work was good.We thank you for your consideration of our revised manuscript, and we would be happy to answer any additional questions.

Response 5: Thanks for your kindly comment.

Round 2

Reviewer 2 Report

Authors carried out changes to the manuscript which improve its comprehension and quality. All the comments were answered technically and sincerely, trying to give clearity to each discussed aspect. I recommend the acceptation for the manuscript.  

Authors carried out changes to the manuscript which improve its comprehension and quality. All the comments were answered technically and sincerely, trying to give clearity to each discussed aspect. I recommend the acceptation for the manuscript.